# Epigenetic Biomarkers of Transition from Metabolically Healthy Obesity to Metabolically Unhealthy Obesity Phenotype: A Prospective Study

**DOI:** 10.3390/ijms221910417

**Published:** 2021-09-27

**Authors:** Carolina Gutiérrez-Repiso, Teresa María Linares-Pineda, Andres Gonzalez-Jimenez, Francisca Aguilar-Lineros, Sergio Valdés, Federico Soriguer, Gemma Rojo-Martínez, Francisco J. Tinahones, Sonsoles Morcillo

**Affiliations:** 1Unidad de Gestión Clínica de Endocrinología y Nutrición del Hospital Virgen de la Victoria, Instituto de Investigación Biomédica de Málaga (IBIMA), 29010 Málaga, Spain; carogure@hotmail.com (C.G.-R.); teresamaria712@gmail.com (T.M.L.-P.); franciscaaguilarjara@gmail.com (F.A.-L.); 2Centro de Investigación Biomédica en Red de Fisiopatología de la Obesidad y la Nutrición (CIBERobn), Instituto de Salud Carlos III, 28029 Madrid, Spain; 3ECAI Bioinformática Instituto de Investigación Biomédica de Málaga (IBIMA), 29010 Málaga, Spain; bioinformatica@ibima.eu; 4Departamento de Endocrinología and Nutrición, Hospital Regional Universitario de Málaga, Instituto de Investigación Biomédica de Málaga (IBIMA), 29009 Málaga, Spain; sergio.valdes@hotmail.es (S.V.); federicosoriguer@gmail.com (F.S.); gemma.rojo.m@gmail.com (G.R.-M.); 5Centro de Investigación Biomédica en Red de Diabetes y Enfermedades Metabólicas Asociadas (CIBERDEM), Instituto de Salud Carlos III, 28029 Madrid, Spain; 6Departamento de Medicina y Dermatología, Universidad de Málaga, 29010 Málaga, Spain

**Keywords:** metabolically healthy obesity, epigenetic biomarkers, metabolic syndrome, DNA methylation

## Abstract

Background: Identifying those parameters that could potentially predict the deterioration of metabolically healthy phenotype is a matter of debate. In this field, epigenetics, in particular DNA methylation deserves special attention. Results: The aim of the present study was to analyze the long-term evolution of methylation patterns in a subset of metabolically healthy subjects in order to search for epigenetic markers that could predict the progression to an unhealthy state. Twenty-six CpG sites were significantly differentially methylated, both at baseline and 11-year follow-up. These sites were related to 19 genes or pseudogenes; a more in-depth analysis of the methylation sites of these genes showed that *CYP2E1* had 50% of the collected CpG sites differently methylated between stable metabolically healthy obesity (MHO) and unstable MHO, followed by *HLA-DRB1* (33%), *ZBTB45* (16%), *HOOK3* (14%), *PLCZ1* (14%), *SLC1A1* (12%), *MUC2* (12%), *ZFPM2* (12.5%) and *HLA-DQB2* (8%). Pathway analysis of the selected 26 CpG sites showed enrichment in pathways linked to th1 and th2 activation, antigen presentation, allograft rejection signals and metabolic processes. Higher methylation levels in the cg20707527 (*ZFPM2*) could have a protective effect against the progression to unstable MHO (OR: 0.21, 95%CI (0.067–0.667), *p* < 0.0001), whilst higher methylation levels in cg11445109 (*CYP2E1*) would increase the progression to MUO; OR: 2.72, 95%CI (1.094–6.796), *p* < 0.0014; respectively). Conclusions: DNA methylation status is associated with the stability/worsening of MHO phenotype. Two potential biomarkers of the transition to an unhealthy state were identified and deserve further investigation (cg20707527 and cg11445109). Moreover, the described differences in methylation could alter immune system-related pathways, highlighting these pathways as therapeutic targets to prevent metabolic deterioration in MHO patients.

## 1. Introduction

Worldwide, obesity has reached epidemic proportions and at least 2.8 million people die each year as result of overweight or obesity. According to the World Health Organization, the prevalence of obesity has nearly tripled since 1975 [1].

Obesity is associated with higher risk of developing metabolic syndrome, type 2 diabetes (T2D) and cardiovascular diseases, resulting in an increase in mortality. However, not all people with obesity present the typical pattern of metabolic complications. This phenotype has been defined as metabolically healthy obesity (MHO) and its prevalence rate varies widely, ranging from 10% to 35% depending on the criteria used and population studied [2,3].

The MHO phenotype can progress to an unhealthy state known as metabolically unhealthy obesity (MUO). It has been suggested that this progression could be a matter of time [4], although there is evidence that suggests that a relevant percentage of MHO individuals maintain their status over time [5].

Despite the growing interest in these groups of subjects, there is a great lack of knowledge concerning the factors that determine why some obese subjects are protected from developing metabolic complications. Different studies propose that higher insulin sensitivity, specific distribution of fat, reduced infiltration of immune cells into adipose tissue, and consequently, a metabolically beneficial cytokine and adipokine secretion pattern, could be some of the mechanisms involved in the genesis of MHO [6,7].

It is estimated the 40–70% of obesity and metabolic disease has a inherited component, but large genome-wide association studies (GWAS) have shown that only 20% of variants in genes related to obesity can explain the predisposition to this condition [8]. Therefore, it has been suggested that epigenetic processes may have a role in the regulation of metabolic diseases. DNA methylation is one of the main epigenetic mechanisms, and can alter gene expression without changing the DNA sequence by adding methyl groups at cytosine residues. This field is still young, but it is attracting interest in various areas such as oncology and metabolic disorders such as obesity.

Previous studies have evaluated the relationship between epigenetic variants and metabolic diseases such as obesity and T2D [9]. It has been suggested that obesity is related to different methylation levels in blood cells compared with those in healthy cohorts [10,11,12]. Also, DNA methylation data from adipose tissue show that epigenetic variation is involved in obesity-associated comorbidities and T2D [13,14].

The aim of the present study was to analyze the long-term evolution of methylation patterns in a subset of MHO subjects in order to search for epigenetic markers that could predict the progression of MHO to MUO.

## 2. Results

Table 1 presents the metabolic variables used to classify the patients included in the study. Briefly, patients were considered as MHO if they had abdominal obesity and <2 of the NCEP ATPIII metabolic syndrome criteria were present. At baseline, triglyceride levels were significantly higher in the unstable MHO group (*p* = 0.001). No statistical differences were found for the rest of the studied variables. At 11-year follow-up, fasting glucose (*p* = 0.01), diastolic (*p* = 0.0159) and systolic blood pressure values (*p* = 0.024) were significantly higher in the unstable MHO group.

### 2.1. Principal Component Analysis

Principal component analysis (PCA) analysis was carried out using the double selection of methylated CpG loci. Firstly, CpG sites that clearly discriminated the two populations at 11-year follow-up were selected; in this selection, both components explained around 58% of the variance (Figure 1A). In this step, a total of 8200 (1%) differentially methylated CpG loci were selected from 815,389 probes, based on component contribution criteria. These CpG sites were tested at baseline to determine markers that discriminated between the two populations both at baseline and at the end of follow-up (Figure 1B). Then, those CpG sites in each component whose contribution values were relevant, were selected. At baseline, sites with a contribution higher than 0.04% (half the maximum contribution value of the best variable) in component 1 and 0.25% (half the maximum contribution) in component 2 were selected and used to establish the methylation changes during the follow-up in the study population.

Finally, 26 significantly differentially methylated CpG sites were selected for further analysis (Appendix A). Most of them (fifteen) were hypermethylated in stable MHO compared to the unstable MHO population both at baseline and 11-year follow-up, while 11 were hypomethylated. The differences between the mean methylation values in both populations at the two study points are shown in Figure 2.

### 2.2. Differentially Methylated Genes

A total of 17 genes and 2 pseudogenes were related to the 26 CpG sites identified in the double PCA selection. The top ten significantly differentially methylated CpG sites were associated with eight unique genes or pseudogenes; two were pseudogenes, namely, nucleolar protein interacting with the FHA domain (*NIPFK3*) and *DTX2P1-UPK3BP1-PMS2P11.* The rest of the sites were unique genes including zinc finger protein, FOG family member 2 (*ZFMP2*), cytochrome P450 family 2 subfamily E member 1 (*CYP2E1*), major histocompatibility complex, class II, DQ beta 1 and beta 2 (*HLA-DQB1* and *HLA-DQB2*), solute carrier family 1 (*SLC1A1*) and phospholipase C zeta 1 (*PLCZ1*). The characteristics of these CpG loci including probe ID, location, gene region or direction of methylation are shown in Table 2.

A more in-depth analysis was performed on these nineteen unique genes (seventeen genes and 2 pseudo genes). All the CpG sites in each of these genes, as well as flanked sequences were collected from the UCSC genome and checked as to whether they are detected in the Methylation EPIC Bead. The CpG sites described in each gene were analyzed to investigate to what extent these genes present multiple different CpG sites in our population.

Fourteen of the nineteen genes identified (73.6%) showed multiple, significant CpG sites. The gene with the largest difference in methylation was *CYP2E1* with 50% of the collected CpG sites differently methylated in the stable MHO and unstable MHO, followed by *HLA-DRB1* (33%), *ZBTB45* (16%), *HOOK3*(14%), *PLCZ1* (14%), *SLC1A1* (12%), *MUC2* (12%), *ZFPM2* (12.5%) and *HLA-DQB2* (8%), and several flanked sequences were identified as differentially methylated in *MUC2*. None of the flanked sequences were found to be significantly differentially methylated in the rest of the genes. The differentially methylated CpG sites in these genes are shown in Appendix A.

### 2.3. Potential Biomarker of Transition to Unhealthy State

A backward stepwise logistic regression was performed using all the methylated sites to evaluate the prediction power of the different methylation in these sites. The final model selected two sites as the best markers to predict the deterioration of stable MHO to an unhealthy phenotype. So, a higher methylation level in the site cg20707527 in the gene *ZFPM2* could have a protective effect against progression to MUO (OR: 0.21, 95%CI (0.067–0.667), *p* < 0.0001); on the contrary, a higher methylation level of the site cg11445109 into the gene *CYP2E1* would increase the progression of the patient to MUO (OR: 2.72, 95%CI (1.094–6.796), *p* < 0.0014). As the baseline triglycerides levels were significantly different, this variable was also included in the model; however, they were not statistically significant.

### 2.4. Enrichment Analysis

The 26 differentially methylated CpG sites selected through double PCA selection were annotated by GO analysis and their functions were classified by biological processes, molecular function, and cellular components using an enrichment analysis. The top 10 GO terms categorized into biological processes, molecular functions and cellular components are illustrated in Appendix A. Biological processes were shown to be linked to the metabolic process of a wide variety of substrates such as halogen compound, benzene, monoterpenoid, etc. Processes not related to metabolism were protein transport, antigen presentation and regulation of cytosolic calcium. Meanwhile, cellular components were mainly associated with transport between membranes, especially Golgi transport or coated-clathrin vesicles (Appendix A).

### 2.5. Pathway Analysis

Finally, pathway analyses were used to assess the biological pathways implicated in the differences between the methylation status in stable MHO and unstable MHO patients related to the 26 CpG sites identified in the double PCA selection. Immune-mediated processes could play a role in the progression to the unhealthy state considering that specific pathways such as Th1 and Th2 activation, antigen presentation, allograft rejection signalling were shown to be hypermethylated in stable MHO (Figure 3).

## 3. Discussion

Identifying those parameters that may predict the metabolic deterioration of MHO phenotype to unhealthy phenotype or the maintenance of metabolic healthy status over the course of time is currently a matter of debate. Among these factors, the role of epigenetics in the stability of MHO phenotype has attracted attention.

DNA methylation represents major epigenetic modification at the transcriptional regulation level. The function of DNA methylation seems to vary with the genomic context (transcriptional start sites, gene bodies, regulatory elements); in this way, DNA methylation of gene promoters is usually associated with transcriptional silencing, while gene body methylation has been associated with transcription enhancement [15].

Previous studies have investigated alterations in DNA methylation in adipose tissue in relation to obesity, insulin resistance and systemic inflammation [16,17], highlighting the relevance of this epigenetic mechanism in obesity and associated comorbidities. Additionally, modifications in the methylation profile of blood cells associated with obesity and metabolic syndrome have been described [18,19]. However, to the best of our knowledge, there is no previous study that evaluates the long-term methylation changes in patients with obesity according to their metabolic status.

Our results showed 26 CpG sites differentially methylated, both at baseline and 11-year follow-up, associated to 19 genes or pseudogenes, which deserve further investigation to decipher their potential role in the stability of MHO phenotype.

Among the pathways altered by these differences in methylation, immune-related pathways stand out as they could be involved in MHO progression to an unhealthy state. It is well-known that obesity is characterized by a chronic low-grade inflammatory state accompanied by macrophage infiltration in adipose tissue. It has been shown that both obesity and T2DM cause dysregulation of the immune system [20,21]. In our population, CpG sites located in *HLA-DRB1* and *HLA-DQB2* genes were shown to be hypermethylated, being higher than the methylation in stable MHO group. These genes belong to the human leukocyte antigen (HLA) class II complex, which is part of the antigen processing and presentation machinery, and a cornerstone of the adaptative immune system. In a previous study, components of HLA class II have shown increased expression in the adipose tissue of patients with obesity and metabolic syndrome [22]. In adipocytes of subjects with obesity, HLA class II has been shown to play a role in triggering inflammation. Indeed, adaptive immunity has been suggested to have a role in the onset and progression of inflammation and insulin resistance in obesity-associated adipose tissue [23]. SNP genotyping has indicated the role of *HLA-DRB1* in T2D [24]. Some *HLA-DRB1* polymorphisms have been suggested to be protective for T2DM; the hypothesized mechanism seems to be a protective role against autoimmune-mediated loss of insulin secretion [25]. Moreover, in obese adolescents, the development of insulin resistance was associated with a down-regulation of *HLA-DRB1* [26].

The rest of the genes associated with the methylation sites described are involved in a wide range of biological processes, highlighting the roles of potential biomarkers that could predict the progression to an unhealthy state at long-term follow-up. Our results showed that higher methylation in cg20707527 (*ZFPM2* gene) and lower methylation in cg11445109 (*CYP2E1* gene) could have a role in the stability of the healthy phenotype in obesity.

In our study, methylation in the *ZFPM2* gene showed a different tendency between groups; our results described two CpG sites that were hypermethylated in stable MHO, whilst in unstable MHO, these CpG sites were hypomethylated at both baseline and 11-year follow-up. *ZFPM2*, also known as *FOG2*, encodes a zinc finger transcription factor that regulates GATA protein activity, including GATA4, which is involved in cardiac function and modulation of angiogenesis [27]; however, it has also been suggested that *FOG2* develops other roles. Previous studies have associated genetic variants of *ZFPM2* with hypercholesterolemia and metabolic syndrome [28,29]. In animal models, triggering inflammation has been shown to lead to a decrease in *FOG2* expression in hepatocytes [30]. In another study, hepatic *FOG2* was shown to attenuate insulin sensitivity by promoting glycogenolysis [31].

The *CYP2E1* gene showed a high proportion of differentially methylated sites, and tended to be hypomethylated in both stable and unstable MHO. Moreover, the hypomethylation levels were higher in stable MHO. *CYP2E1* belongs to the superfamily of enzymes, cytochrome P450 (CYP), whose members are involved in the biotransformation of drugs, xenobiotics and endogenous substances [32]. The increased activity of *CYP2E1* may promote oxidative stress due to its ability to produce excessive reactive oxygen species [33]. This induction has been described at hepatic level in patients with non-alcoholic fatty liver disease [34]. Additionally, *CYP2E1* activity has been shown to be higher in patients with obesity [35] and an animal model of metabolic syndrome [36]. Although the results are contradictory, some studies have suggested an increase in *CYP2E1* activity in patients with T2D [37], and both glucose and insulin may modulate its activity [38]. All these data suggest that *CYP2E1* may have a role in metabolic alterations with an inflammatory component.

To the best of our knowledge, this is the first study to perform a longitudinal analysis of methylation status in an obese population with a 11-year follow-up. However, our study also presents some limitations. We used blood samples to assess differential DNA methylation, therefore further research on tissue-specific methylation patterns would be necessary. We could not perform RNA analysis to relate DNA methylation to gene expression. Due to the sample size, some relevant differences may not have been detected. Additionally, although the Infinium EPIC array is a very useful tool to interrogate CpGs sites, it only covers 30% of the human methylome. Finally, a validation cohort would be necessary to confirm our results although obtaining a cohort for long-term follow-up (11 years) makes the validation overly complicated.

For a better understanding of the MHO phenotype as well as the predictors factors in the transition of MHO to MUO, more longitudinal studies with a larger number of subjects will be needed. Epigenome-wide studies with samples from adipose tissue will be required in order to increase our knowledge of the mechanisms involved in the development of the MUO phenotype.

## 4. Materials and Methods

### 4.1. Design and Subjects

This study is part of the Pizarra study, the details of which have been previously published [4,39]. Briefly, the Pizarra study is a prospective, population-based cohort study of 1051 subjects aged 18–65 years from Pizarra, a town in the province of Malaga (Andalusia, southern Spain). The cohort was re-evaluated after 11 years, and a total of 547 individuals completed the follow-up. Blood samples at both baseline and 11-year follow-up were available from 276 of 547 individuals who completed the follow-up. Of them, 137 patients were obese, both at baseline and 11-year follow-up. Among 137 patients, 58 were classified as MHO at baseline. After matching by age, 18 patients were selected to be included in the study.

Informed consent was obtained from each participant, and the study was approved by the medical ethics committee of the Carlos Haya Regional University Hospital of Malaga.

### 4.2. Classification Criteria

The NCEP ATPIII criteria were used to classify the subjects according to their metabolic status [40]. They were considered as MHO if they had abdominal obesity (waist circumference >102 cm in men and >88 cm in women) and <2 of the NCEP ATPIII metabolic syndrome criteria were present: systolic blood pressure ≥135 mmHg or diastolic blood pressure ≥85 mmHg; fasting plasma glucose concentration ≥100 mg/dL; HDL-C concentration <40 mg/dL in men and <50 mg/dL in women; fasting plasma TG concentration ≥150 mg/dL; or treatment with antihypertensive, lipid lowering, or glucose-lowering medications.

For this study, a subset of 18 MHO subjects at baseline were selected for genome-wide DNA methylation analysis. Of these, 9 MHO subjects developed metabolic complications at 11-year follow-up (unstable MHO; n = 9), whilst the other sub-set of samples remained metabolically healthy at 11-year follow-up (stable MHO; n = 9).

### 4.3. Procedures

Weight and height measurements were made at baseline and 11-year follow-up. Body mass index (BMI) was calculated as: weight (kg)/height^2^ (m^2^). Blood pressure was measured twice with a sphygmomanometer with an interval of 5 min between measurements and the average of the two measurements was used in the analyses.

At baseline and 11-year follow-up, blood samples were collected after a 10–12 h fast. The serum was separated, and blood and serum samples were immediately frozen at −80 °C until analysis. Biochemical variables were measured in duplicate. Blood glucose was measured using the glucose oxidase method (Bayer, Leverkusen, Germany). Enzymatic methods were used to measure total cholesterol, triglycerides, and high-density lipoprotein cholesterol.

### 4.4. DNA Methylation Assay

DNA was extracted from peripheral blood using the QIAmp DNA Blood Mini Kit (Qiagen, Hilden, Germany) following the manufacturer’s instructions. DNA concentration was quantified with a Qubit 3.0 Fluorometer (Thermo Fisher Scientific, Waltham, MA, USA) using Qubit dsDNA HS Assay Kit Fluorometer (Thermo Fisher Scientific, Waltham, MA, USA) After quantification, a total of 500 ng of genomic DNA was bisulfite-treated using a Zymo EZ-96 DNA Methylation™ Kit (Zymo Research Corp, Irvine, CA, USA) and was purified using a DNA-Clean-Up Kit (Zymo Research Corp, Irvine, CA, USA).

Over 850,000 methylation sites were interrogated with the Infinium Methylation EPIC Bead Chip Kit (Illumina, San Diego, CA, USA) following the Infinium HD Assay Methylation protocol, and raw data were obtained from iS (Illumina) software.

### 4.5. Methylation Data Analysis

We used statistical programming language R 3.5.1 (https://www.r-project.org/, accessed on 1 April 2021) to perform the methylation data analysis. Raw data files (idat files) were read with the minfi package [41] to calculate raw β-values. Normal-exponential out-of-band (NOOB) normalization [42] was used to correct the background. Probes located at sexual chromosomes or near SNPs were removed from the analysis. Low quality probes (those with a detection *p*-value > 0.01 in at least 10% of samples) were also removed. Finally, beta-mixture quantile (BMIQ) normalization [43] was applied to correct for the two different bead designs in the microarrays. For the differential methylation analysis, we transformed β-values to M-values.

### 4.6. Statistical Analysis

Statistical analysis and comparison were performed using R software (3.5.1) to study differences in anthropometric and biochemical variables with the Kruskall–Wallis test for continuous data and the chi-square test for categorial data. Data are expressed as the mean ± standard deviation, or as a percentage. Values were statistically significant when *p* < 0.05.

#### Principal Component Analysis (PCA)

Two complete datasets of normalized CpGs sites were obtained at baseline and the 11-year follow-up. Principal component analysis (PCA) was implemented using native R implementation through R Studio Software 1.2.5033 (version 3.5.1). Classical PCA can be considered as a projection-based approach to find the low-dimensional space that best represents a cloud of high-dimensional points [44]. Firstly, we performed PCA on the dataset of the 11-year follow-up and used the most important CpG sites in both components as subsets for the dataset. Around 1% of them were selected (8200 CpG sites). These sites were tested at baseline and those with a contribution higher than 0.04% (half the maximum contribution value of the best variable) in component 1 and 0.25% in component 2 were selected and used to establish the methylation changes through the follow-up of the study population.

To validate the importance of the selected CpG sites in the PCA, a comparative analysis was performed for each site. Differences between groups were established by using the Kruskal-Wallis test.

Differentially methylated CpG sites identified at both baseline and 11-year follow-up were used to perform a backward stepwise logistic regression to evaluate the prediction power of these sites for the progression to a metabolically unhealthy obesity state.

### 4.7. Gene Ontology and Pathway Testing

These CpG sites were studied using two different approaches; on one hand, Gene ontology (GO) was used to determine the main processes associated with the selected CpG sites by using AmiGO, a web application that allows users to query, browse and visualize ontologies and related gene product annotation (association) [45]. On the other hand, the selected CpG sites were analyzed through ingenuity pathway analysis software from QIAGEN. This software allowed us to determine which canonical pathways were related with the selected CpG sites and to establish the more relevant processes altered in both groups at the follow-up. Finally, statistical analysis was performed using R software (3.5.1 version).

## 5. Conclusions

In conclusion, we described differentially methylated sites that could have a role in the stability/worsening of MHO phenotype; among them, two potential biomarkers have been suggested (cg20707527 and cg11445109). Moreover, the described differences in methylation could alter immune system-related pathways, suggesting these pathways as therapeutic targets to ameliorate metabolic deterioration in MHO patients.

## Figures and Tables

**Figure 1 ijms-22-10417-f001:**
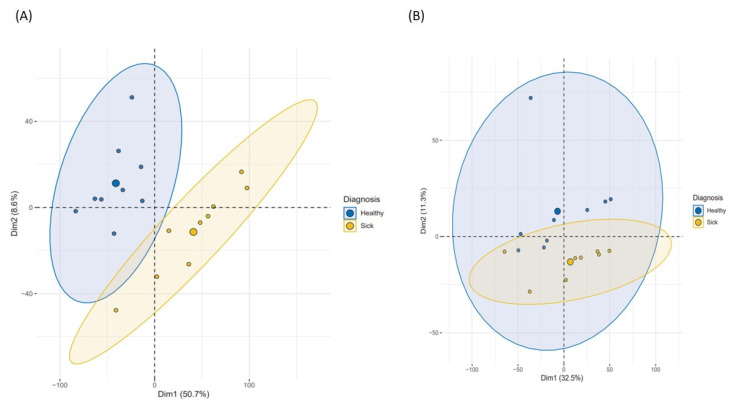
PCA at baseline and 11-year follow-up. (**A**) PCA performed on 11-year follow-up dataset. (**B**) PCA of the most important 11-year follow-up methylation sites at baseline.

**Figure 2 ijms-22-10417-f002:**
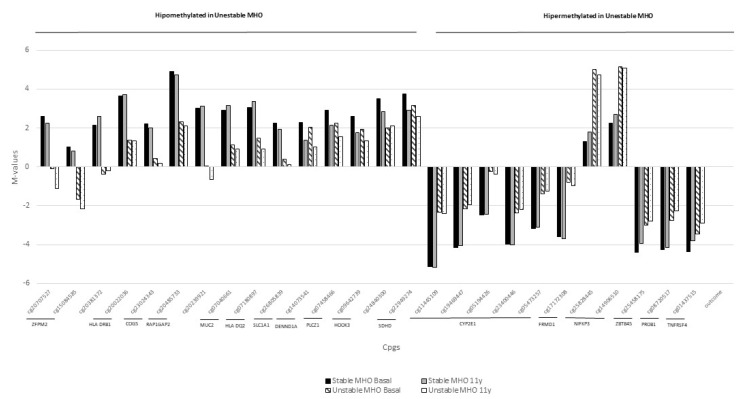
Methylation levels of 26 significantly differentially methylated CpG sites identified in the double PCA selection at baseline and 11-year follow-up.

**Figure 3 ijms-22-10417-f003:**
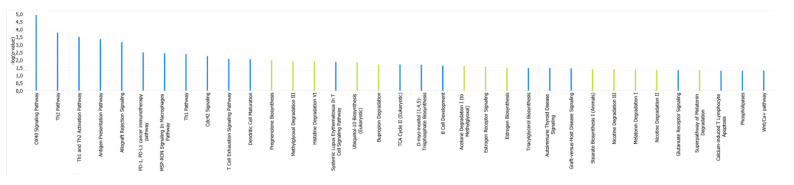
Main pathways related to the significantly differentially methylated CpG sites. Blue bars: biochemical pathways with CpGs sites hypermethylated in stable MHO. Green bars: biochemical pathways with CpGs sites hypermethylated in unstable MHO.A network analysis was performed to examine the inter-relationships between these genes. Almost half of them were linked in a unique network with transcription factors and transcription regulators (AHR, SIP1 or HNF4A) as the main nodes (Figure 4).

**Figure 4 ijms-22-10417-f004:**
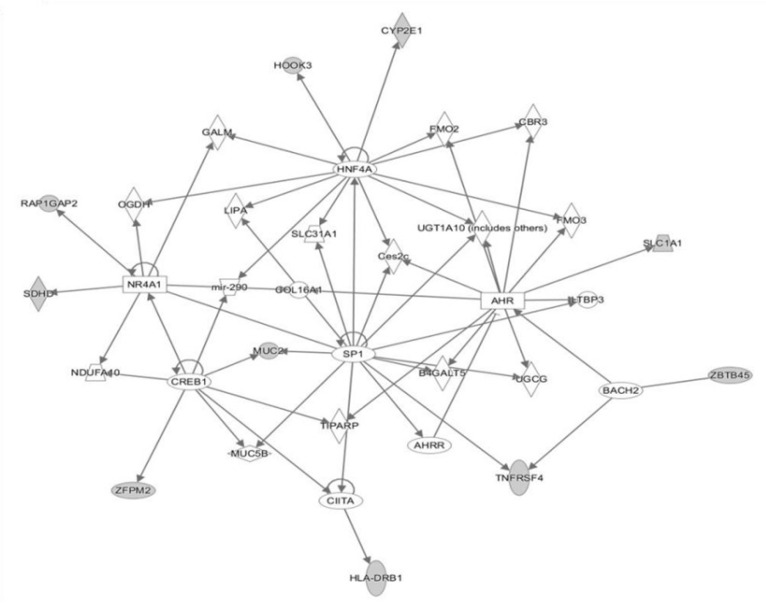
Gene network of interactions related to more relevant CPGs sites.

**Table 1 ijms-22-10417-t001:** Anthropometric and biochemical characteristics of the subjects included in the study.

Title 1	Baseline			11-Year Follow-Up		
	Stable MHO (n = 9)	Unstable MHO (n = 9)	*p*-Value	Stable MHO (n = 9)	Unstable MHO (n = 9)	*p*-Value
Age	45 ± 11	53 ± 9	NS			
Gender (Male/Female)	2/7	3/6	NS			
Fasting glucose (mg/dl)	103.5 ± 13.3	106.6 ± 10.3	NS	90 ± 4.9	108.4 ± 16.6	0.01
BMI	28.2 ± 1.6	29 ± 4.3	NS	29.9 ± 3.4	31.1 ± 4.0	NS
Triglycerides (mg/dl)	52.7 ± 12.3	92.7 ± 37.8	0.01	82 ± 25.2	100.33 ± 53.0	NS
HDL-cholesterol	57.8 ± 12.3	51.4 ± 9.6	NS	60.7 ± 6.3	53.6 ± 8.4	0.06
DBP (mm Hg)	81.4 ± 7.5	88.6 ± 16.8	NS	75.7 ± 9.3	90.5 ± 11.5	0.015
SBP (mm Hg)	121 ± 16.9	138.6 ± 26.3	NS	126.3 ± 19.3	153.8 ± 23.2	0.024
HTA treatment (%)	0	22.2	NS	0	55.6	0.015

Data are expressed as the mean ± standard deviation, or as (percentage). *p*-values for continuous data were calculated using the Kruskal–Wallis test, and for categorical data they were calculated using the chi-square test or Fisher’s exact test if the frequency was <5. BMI: body mass index. HDL cholesterol: high density lipoprotein cholesterol. DBP: diastolic blood pressure. SBP: systolic blood pressure. HTA treatment: arterial hypertension treatment.

**Table 2 ijms-22-10417-t002:** The top ten significantly differentially methylated CpG sites in stable MHO and unstable MHO throughout the study.

Probe ID	Location	Gene Symbol	Gene Name	*p*-Value	Hypermethylated
cg20707527	Chr: 8 q23.1	*ZFPM2*	Zinc Finger Protein. FOG Family Member 2	0.0001	Stable MHO
cg15084585	Chr: 8 q23.1	*ZFPM2*	Zinc Finger Protein. FOG Family Member 2	0.0001	Stable MHO
cg20022036	Chr: 6 p21.32	*HLA-DRB1*	Major Histocompatibility Complex. Class II. DR Beta 1	0.0015	Stable MHO
cg20239921	Chr: 7 q 11.23	*DTX2P1-UPK3BP1-PMS2P11*	DTX2P1-UPK3BP1-PMS2P11	0.0015	Stable MHO
cg26805839	Chr: 9p24.2	*SLC1A1*	Solute Carrier Family 1 Member 1	0.0035	Stable MHO
cg11445109	Chr: 10 q26.3	*CYP2E1*	Cytochrome P450 Family 2 Subfamily E Member 1	0.0046	Unstable MHO
Cg07180987	Chr: 6 p21.32	*HLA-DQB2*	Major Histocompatibility Complex. Class II. DQ Beta 2	0.0057	Stable MHO
cg05194426	Chr: 10 q26.3	*CYP2E1*	Cytochrome P450 Family 2 Subfamily E Member 1	0.0057	Unstable MHO
cg25828445	Chr: 12 p13.31	*NIFKP3*	Nucleolar Protein Interacting with The FHA Domain	0.0067	Unstable MHO
cg07458466	Chr: 12 p 12.2	*PLCZ1*	Phospholipase C Zeta 1	0.0083	Stable MHO

## Data Availability

The datasets analyzed during the current study will be available on the GEO platform when the manuscript is accepted. Epigenome data from the Infinium Methylation EPIC Bead Chip and metadata including identifier and group of patients will be deposited.

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
