# Peer review of "Epigenetic Biomarkers of Transition from Metabolically Healthy Obesity to Metabolically Unhealthy Obesity Phenotype: A Prospective Study"

_ijms, 2021, doi:10.3390/ijms221910417_

Round 1

Reviewer 1 Report

The paper is worth to be published since it adds interesting material to scientific discussion in this field. At the same time, I have a few suggestions. First authors should add a section with limitations of the study. For instance, only 18 patients were selected to be included in the study, it is well known that if the study is too small you may miss important differences (because of chance variation). Moreover, only DNA methylation data from blood samples were analyzed, while obesity is characterized by a chronic low-grade inflammatory state, accompanied by macrophage infiltration in adipose tissue. In addition, what about specific distribution of fat, could this affect the transition of metabolically healthy obesity to metabolically unhealthy obesity.

Additionally, researchers analyzed only 850 000 methylation sites (Infinium Methylation EPIC Bead Chip Kit (Illumina, San Diego, CA, USA), however, the EPIC array covers only 30% of the human methylome.

Also it is better to add a section suggest specific future avenues of research.

Author Response

We are grateful for the comments and suggestions from the reviewer. We are going to reply point by point all the questions.

  • Regarding the limitations of the study, we have included some additional limitations such as the capacity of the Infinium Methylation EPIC Bead Chip Kit, and the possibility that some differences have been missed due to the small sample size. Anyway, this comment is even positive because highlights the importance of the differences found.
  • We are conscious that we have only analyzed data from blood samples and this limitation was also included in the discussion. As the reviewer comments, obesity is characterized by a chronic low-grade inflammatory state, accompanied by macrophage infiltration in adipose tissue. Unfortunately, we don’t have samples from adipose tissue of these subjects to analyze the epigenetic pattern in this tissue.
  • Regarding the specific distribution of fat, in our study, the selection of the patients was based on abdominal obesity (NCEP ATPIII criteria), a relevant component of the metabolic syndrome, independent of overall obesity. Furthermore, there weren’t significant differences in this parameter between both groups, baseline and at follow-up.
  • Finally, as the reviewer suggests, we have added a final paragraph suggesting future avenues of research.
  • All the changes have been highlighted in the text using the Track Changes function.

Reviewer 2 Report

There is a very interesting paper presenting changes in methylation profile during long follow-up.

However, some improvement is needed before this paper will be published in a top scientific journal.

An editorial work is still needed.

At the beginning of the result-section it is necessary to underline how MHO and MUO subjects were recognized. Including this information only under Methods  is not sufficient for the clarity of the description of the results.

Section 2.1 – in the title of this subsection, the abbreviation PCA should not be used, but its full expansion

Fog 3 legend should be modify as “biochemical pathways” are not “methylated”

Based on presented data it is hard to recognized how large  differences in methylation levels of specific genes were observed among MHO and among MHU subjects. It is of importance, especially that each group consisted of 9 subjects.

Author Response

Thank you for the comments.

We have done all the changes suggested by the reviewer.

  • Now we have included, in the results section, how both phenotypes were classified.
  • The abbreviation PCA has been removed from the title of the Section 2.1
  • Figure legend 3 has been modified.
  • Lastly, the reviewer asks about the difference in methylation levels of the specific genes. To this respect, we have shown the Methylation levels (M-value) for the 26 differentially methylated CpGs sites. In this graph (Figure 2) we can observe the changes between both phenotypes along the time, as well as the magnitude of the change. Furthermore, we explored in depth these 19 genes and observed that 14 of them showed multiple significant CpGs differentially methylated. This information is found in Additional File 2. It is important to keep in mind that we are analyzing specific CpGs sites in a gene, but we cannot guarantee that the same methylation profile will be conserved throughout the gene.
  • All the changes have been highlighted in the text using the Track Changes function.
  • English language has been checked and some changes have been done.